



# A clustering-based method for identifying and tracking squall line

Zhao Shi[1,2], Yuxiang Wen[1,2], Jianxin He[1,2]

[1]College of Atmospheric Sounding, Chengdu University of Information Technology, Chengdu 610225, China
[2]Key Laboratory of Atmospheric Sounding, China Meteorological Administration, Chengdu 610225, China

*Correspondence to*: Yuxiang Wen (3220305009@stu.cuit.edu.cn)

**Abstract.** The squall line is a type of convective system characterized by storm cells arranged in a line or band pattern, which is usually associated with disastrous weather. The identification and tracking of squall lines plays an important role in early

warning of meteorological disasters. Based on weather radar data, a clustering-based identifying and tracking algorithm for squall lines is presented. Clustering analysis is designed to distinguish the strong echo area and estimate the feature value including reflectivity value, length, width, area, endpoints, central axes, and centroid. The clusters in the linearly arranged form are merged to improve the identification ability in the development stage of squall lines. The three-dimensional structure and movement tracking of the squall line are obtained using the centroid and velocity of the squall lines identified in a single

layer. It is demonstrated that the method can effectively identify and track one or more squall lines over a weather radar scanning area. The result shows that the recognition accuracy rate in the single scan elevation of this method is 95.06%, as well as a false positive rate of 3.17%. This method improves the accuracy of squall line identification in the development stage of squall lines and works relatively efficiently even when high interference contaminates.
.

**1 Introduction**

The squall line is a prevalent convective weather commonly occurring in mid-latitude regions during spring and summer. Squall lines are arranged by storm cells in a linear structure, which can extend over one hundred or even hundreds of kilometers. The typical life cycle of a squall line is about 6-12 hours (Rotunno et al., 1988; Wanghong et al., 2009). The squall lines occurring in coastal areas can bring large amounts of precipitation inland from coastal areas (Oliveira and Oyama, 2020).

Squall lines are also associated with severe disastrous weather events, including heavy precipitation in brief periods, lightning, hail, downbursts, and even tornadoes (Trapp et al., 2005; Xiaohong et al., 2021). Therefore, identifying and tracking squall lines are crucial for early warning of meteorological disasters.

The weather radar is an effective meteorological remote sensing instrument with high spatiotemporal resolution, and it has been widely used in monitoring and nowcasting of mesoscale convective systems. To enhance radar meteorology

understanding on squall lines, the NOAA has conducted many studies on squall lines using Doppler weather radar data in the last century (Smull and Houze, 1985; Srivastava et al., 1986; Smull and Houze, 1987; Bluestein et al., 1987). The evolution



mechanism of the squall line is analyzed with radar observation. Combining the Vertical Integrated Liquid water (VIL), Echo Top (ET), composite reflectivity, and VAP calculated wind field data and auto-weather station observation data, the relationship between squall lines and short-term heavy rainfall, strong winds, hail, and other disastrous weather processes is revealed. The occurrence of squall lines may lead to clockwise vertical wind shear at low altitudes and counterclockwise vertical wind shear at high altitudes. This shear favored the generation and strengthening of unstable weather and provided a favorable environment for the development of convection. At the same time, the dramatically changing VIL and high ET often heralded hail and strong wind (Wanghong et al., 2009).

One of the characteristics of squall lines in weather radar data is the formation of a strong echo band on the radar reflectivity image (Ma, 2022), making it visually identifiable. However, their suddenness, and wide-ranging impact make it hard to improve real-time forecasting capabilities by manual identification. Meanwhile, the automatic identification of squall lines is a complex task (Chengling et al., 2017). Over the past few decades, numerous researchers have conducted studies in this field and have employed various algorithms. such as the two-dimensional Fourier transform (Kelly, 2003), wavelet transform, Hu moment theory (Chengling et al., 2017), Hough transform (Wang et al., 2021; Chengling et al., 2017), and machine learning (Ziqi et al., 2021).

Rinehart and Garvey used traditional radar, based on the cross-correlation method, Fourier analysis, and Gaussian curve fitting to detect the movement of the storm, and the process of merging and splitting (Rinehart and Garvey, 1978), Kelly et al. proposed a method in their patent using Two-dimensional Fourier transform and morphological methods to build a system to identify the radar image features that may accompany hazardous weather (Kelly, 2003), and draws a visual boundary around the features to emphasis them while displaying the radar image to the users. "SCIT" - storm unit identification and tracking algorithm (Johnson et al., 1998), using centroid tracking technology to identify and track the individual storms, and provide storm characteristic information. It can identify storms (including isolated storms, cluster storms, supercells, and squall lines), can correctly track more than 90% of storm cells, and has been applied in the WSR88d radar system. The 'TITAN' algorithm (Dixon and Wiener, 1993) defines a "storm" as a continuous area that exceeds the reflectivity and size thresholds (adjacent areas with a reflectivity exceeding 35dbz and a volume exceeding $50km^3$). Storms are tracked using the results of the comparison of the previous scan data with subsequent scans with the storm's movement characteristics and maximum horizontal movement speed. Morphological methods were used to deal with the merging and splitting of storms.

Promoting the technological development of automatic identification, tracking and prediction of severe weather is a long-pursued research topic. In recent years, weather radar networks have been widely deployed in densely populated areas around the world for severe weather monitoring and early warning, and radar meteorological data has explosively increased. With the development of digital image processing, big data mining, artificial intelligence, and other technologies, the bad weather automatic recognition algorithm has presented a new development trend. Due to the length and the large area of the squall line, as well as the special arrangement (especially when the squall lines are arranged in a shape like 'L'), identifying the squall line using the methods similar to storm cells will result in a lower identification accuracy (Gangqiang et al., 2021). Convolutional neural network (CNN) is used in squall line identifying (Ziqi et al., 2021). The proposed model effectively identifies the



presence of squall lines during the early development stage and the mature stage of convective, even when the reflectivity value is lower than the exuberant stage. However, the amount of dataset used for training the model is limited, which may lead to false positives and false negatives. The maximal margin detection method based on wavelet transform patterns and the Hu moment principle (Chengling et al., 2017) can extract the echo characteristics of squall lines, specifically the exuberant stage

when the storm cells form in a connected line. The squall line recognition algorithm based on wavelet transform and Hu moment theory can determine whether a particular time frame corresponds to the occurrence or dissipation of a squall line. The presence of squall lines in radar echoes is determined using a threshold value. The selection of the threshold value relies on extensive experimental data or specific requirements. If the intensity of the radar echo falls below the threshold, it can be identified as a squall line. This approach facilitated automated squall line recognition from radar echoes and achieved effective

identification of squall lines. However, using a limited number of thresholds may result in false positives and false negatives. The clustering algorithm is a kind of algorithm commonly used in machine learning and data mining (Gower, 1967), and it is also a kind of unsupervised learning algorithm. In the algorithm of using supervised machine learning to identify squall lines, it is necessary to label the existing data in advance. However, the appearance of squall lines is random in time and space, making the data labeling a complicated project. The clustering algorithm can classify the data points by using some

characteristics of the data points without pre-setting the labels, which is very effective for the squall line process with strong randomness. The clustering algorithm can be used to classify the points in the radar scan results, and the data points can be classified based on certain characteristics (such as distance or density). Each individual scan result of the radar sample can correspond to the Euclidean space. The clustering classifies these points into multiple irrelevant sets marks them according to certain information, and uses other features of the set to identify the weather system. When used in conjunction with squall

line features, the clustering algorithm can identify clusters that meet certain criteria in the radar reflectivity factor data, and extract data points that are associated with squall lines.

## 2 Materials and Methods

The method mainly comprises the following processing: Data Preprocessing, Threshold calculation, Clustering analysis, Target identifying, and Target tracking.

### 2.1 Data sources

China and the United States are both countries significantly impacted by natural meteorological disasters. It is crucial for both nations to undertake meteorological observation, and weather prediction, and facilitate scientific assistance in disaster prevention, mitigation, and response to climate change. Over the years, the meteorological agencies of both countries have collaborated extensively in fields such as meteorological observation and forecasting operations, sub-seasonal forecasting

product research and development for operational usage, and investigating major climatic phenomena. In 1996, China started to develop a new generation weather radar named as CINRAD. This paved the way for a new generation of weather radars to





be installed across the country. Meanwhile, both China and the US boast expansive land areas, comprehensive weather radar networks, and rich weather observation data. The method is based on volume scan data from NEXRAD and CINRAD. The method is based on the weather radars that are already in operational usage. These radars were well calibrated, and the data

obtained by these radars were preliminarily quality-controlled, including ground clutter suppression, velocity de-aliasing, attenuation revision, and so on. These data have been in use for years and have proven to be reliable most of the time.

**2.2 Data Preprocessing**

Radar volume scan data is stored in the form of polar coordinates (azimuth and radial distance). In order to facilitate spatial correspondence and further processing, the data needs to be converted to cartesian coordinates and interpolated by the nearest

neighbour method into a regular spatial grid. The distribution of radar echo information in real space is calculated using elevation($(El)$), azimuth($Az$) and radial distance($r$).

$$x = \sin(Az)\sin(El)r \; ; \; y = \cos(Az)\sin(El)r \qquad (1)$$

Finally, the nearest-neighbor interpolation is applied to interpolate the spatial points into the corresponding grid, enabling the gridded data to reflect the actual weather conditions from weather radar scanning.

**2.3 Threshold calculation**

One of the most important features observed in squall lines is that there are large areas in the girded radar echo image that have significantly higher reflectivity values than the surrounding area, typically ranging from 40 to 52dBZ (Wang et al., 2019). In previous studies, one of the approaches to extract these points is using a threshold to extract these points, for which various threshold settings have been proposed by previous researchers:

| Num | Threshold settings |
| --- | --- |
| 1 | The length of echo bands with reflectivity greater than 12dBZ should be no less than 150 km, and the length-to-width ratio of bands with reflectivity greater than 36dBZ should be no less than 3:1 (Chen and Chou, 1993). |
| 2 | The length of echo bands with reflectivity greater than 20dBZ should be no less than 100 km, and the length-to-width ratio of bands with reflectivity greater than 40dBZ should be no less than 5:1 (Geerts, 1998) |
| 3 | The length of echo bands with reflectivity greater than 40dBZ should be no less than 100 km (Parker and Johnson, 2000) |

**Table 1. Threshold settings of the previous studies**

However, different threshold settings will lead to different identifying results. Without proper threshold settings, the method may output wrong identifying results. So, a reasonable selection of thresholds is necessary. The thresholds are mainly focused





on the following parts: The reflectivity value threshold to extract storms from weather radar data, and the length and width threshold to identify the storm as squall lines.

Combined with the study on South China which used raindrop spectra combined with polarimetric radar (Wang et al., 2019), the threshold conditions are set as follows in this method:

**Table 2. Threshold settings of this research**

| Num | Threshold settings |
|---|---|
| Threshold 1 | The minimum values of the reflectivity in the region are not less than 40dBZ. |
| Threshold 2 | The maximum values of the reflectivity in the region are not less than 52dBZ. |
| Threshold 3 | The length of the region is not less than 100 kilometres. |
| Threshold 4 | Length to width ratio of the region not less than 3:1 |

### 2.4 Clustering analyzing

The data points that satisfy threshold 1 are extracted using the minimum reflectivity threshold, and it should be noted that the regions extracted using this threshold may contain points that do not belong to squall lines, such as storm cells, noises, and clutter points. Therefore, further analysis of these points is required to differentiate squall lines and other points. The main steps of the cluster analysis process include Region clustering, Region characterization, and Region combination.

### 2.4.1 Region clustering

In this step, a clustering method based on points coordinates, density, and searching distance features is proposed to classify the points extracted by threshold 1. Three parameters are needed in region clustering (shown in Figure 1, the distances mentioned in this method are refer to the Euclidean distance): the radius of the field (Eps) and the minimum number of points required to judge the core points (MinPts) and the search condition to form clusters (Searching distance). In this method points in radar data are categorized into core points and noise points by whether or not the core conditions are met. The core condition means that there are at least MinPts points within the Eps distance of the point itself. Points that satisfy the core condition are core points, and vice versa are noise points. If there is a series of core points, and the distance between each core point is within the searching distance, these core points are assigned to the same cluster. The method iterates over all the points in the extracted region above. A set of clusters and a composition of unclassified noise points are finally obtained. This method improves the classification ability of the method, especially in the condition that the squall line identification process and the line-arranged convective cells are not fully merged or when there is occlusion or interference in radar data itself.





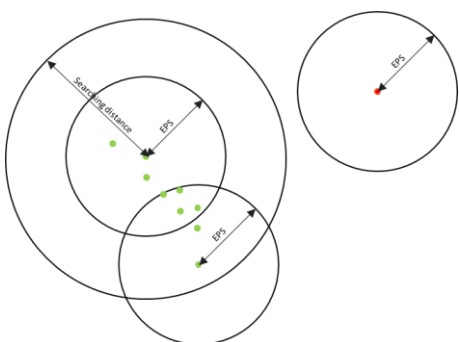

**Figure 1 Region clustering parameters**

Detailed steps to realize the Region clustering are as follows:

Parameter Selection: Determine the three parameters: Eps, MinPts, and Searching Distance.

Initialization of labels: Initialize the clustering labels by assigning labels to all points as 'not labeled'.

Core point identification: For the point labeled as 'not labeled', determine whether it meets the core condition, and if it meets

the core point condition, mark it as a core point, and classify it into the cluster of the current label, and vice versa are marked as noise points.

Cluster formation: For other points within the searching distance of the current core point, determine whether they meet the core condition, and if so, mark them as the core point, and classify them into the current clustering, and vice versa are marked as noise points. Repeat this step until all points within the searching distance are labeled.

Clustering iteration: When there is no core point in the searching distance assigned as 'not labeled', update the label to traverse the 'not labeled' points without the searching distance, and repeat the above steps until all points are labeled.

Output results: output point coordinates and corresponding labels.

**2.4.2 Region characterization**

Based on the region clustering results obtained above, further analysis of the features of the clusters is required. The features

include the central axis, the endpoints, the area, the intensity (maximum reflectivity value), the velocity, and the position of the centroid.

The clusters' velocity, intensity, and area can be easily obtained by spatial transformation and correspondence. However, determining the central axis and endpoints of the clusters is difficult because the weather systems are unstable, and this results in an irregular shape of the clusters. Therefore, it is necessary to find efficient and accurate methods to estimate the central

axis of the clusters. The Hough transform is an image processing algorithm published by Hough et al. in 1959 (Hough, 1959), which has been widely used to recognize lines or circles in complex images (Duda and Hart, 1972), and it has also been used in squall line identifying in previous research (Wang et al., 2021). The Hough transform is a method that utilizes a voting-based approach to transform a collection of lines into a collection of points. It transforms the point space (X, Y) into parameter



space $(\rho, \theta)$ to form a series of voting accumulations. The parameter space consists of two parameters: $\rho$ and $\theta$. The point

coordinates X and Y are converted to $(\rho, \theta)$ by the following equation:

$$\rho = X * \cos(\theta) + Y * \sin(\theta) \tag{2}$$

Hough peak is the partial maximum in the point set voting results. These partial peaks correspond to the most voted parameter

combinations $(\rho, \theta)$ and represent potential lines in the original data. The set of parameters $(\rho, \theta)$ of Hough peak is associated

with a straight line in the gridded radar data. The $(\rho, \theta)$ obtained at this point corresponds to straight lines in gridded radar

data (as shown in Figure 2), and this straight line can be considered as central axes for clustering.

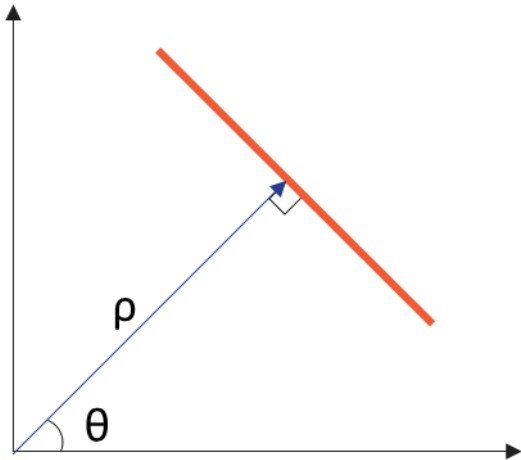

**Figure 2 Correspondence between straight line and parameter$(\rho, \theta)$**

Meanwhile, the straight line intersects with the edge of the clusters, these intersection points are considered to be the endpoints

of the clusters, and the length of the central axis can be estimated using the endpoints. The clusters associated with the storms

are approximately considered as ellipses, and the width of the clusters are calculated using the area and the length of the central

axes obtained from the above steps:

$$width(SQL) = \frac{4 * aera}{\pi * length(SQL)} \tag{3}$$

In this formula, the $aera$ is the area of the cluster, and $length$ is the estimated length of the cluster.

### 2.4.3 Region combination

In order to enable the algorithm to identify the squall lines before they are fully formed (storm cells that are not fully merged

at the development stage of the squall line but are linearly arranged), in this step the clusters that are linearly arranged will be

merged. The traditional method uses the centroid distance to determine whether to merge the storms. However, the squall line,

due to the special characteristics of its linear arrangement and large length-to-width ratio, the judging method that searching

simply by the distance between centroids in accordance with the distance circle has some drawbacks, when the distance is

taken to be a large value, the non-linearly-arranged clusters on the 'width' direction (which does not line up with other clusters





but within the distance) may be merged, and when the value is taken to be a small value, the clusters on the 'Length' direction are unable to be searched.

This algorithm determines whether to merge two clusters by the distances between the obtained endpoints above. If the two clusters' endpoints are within 10km, then the two clusters are combined into the same cluster. After the two clusters are merged, the features will change, so the re-calculation of the features is needed. The length and area of the clusters are added together as the area and length of the newly merged clusters, the maximum reflectivity factor, and the width are taken to be the larger value of the two clusters. The reflectivity value is considered as 'mass', and the coordinates of the centroid in the horizontal direction are calculated as follows:

$$X = \frac{\sum j * Zh(i,j)}{\sum Zh(i,j)}, Y = \frac{\sum i * Zh(i,j)}{\sum Zh(i,j)} \tag{4}$$

**2.5 Target identifying**

The cluster analysis results obtained from the cluster analysis step are compared with the threshold conditions 2-4 (Mentioned in Table 2), and if there is a cluster that meets the threshold, it means that at least one squall line exists in this layer of the current volume scanning data, and this cluster is considered to be an identified squall line.

**2.6 Target Tracking**

The above steps enable the method to identify the squall lines and obtain the location of squall lines in a single layer of volume-scan data. However, in practice, the vertical structure plays a more important role than the horizontal structure during strong convective weather that is prone to cause major meteorological disasters (Ma, 2022). Therefore, it is necessary to obtain the three-dimensional structure of the squall lines' radar-scanned information. At the same time, convective storms are characterized by rapid structural evolution and movement. Over the life cycle of the squall line, it may undergo multiple splits, regenerations, and reorganizations (Ye-Qing et al., 2008). So it is also necessary to track the changes in the squall lines' shape and location in practical applications.

The wind field and velocity information are calculated by the VAP method from radar radial velocity data. The traditional VAP method assumes that the wind field is uniform in the region, and the calculating ability is good in the case of a uniform wind field, but the error is larger in the case of a non-uniform wind field, so the extended-VAP(EVAP) (Zhouzhenbo et al., 2006) is used to calculate the wind field. The EVAP inversion method is as follows:

$$\cos(\Delta\theta + \Delta\beta) = \frac{V_{r1} + V_{r2}}{V_r} \tag{5}$$

$$\tan\beta = (\frac{V_{r1} - V_{r2}}{V_{r1} + V_{r2}})\cot(\Delta\theta + \Delta\beta) \tag{6}$$

$$V = \left|\frac{V_r}{\cos\beta}\right| \tag{7}$$





where $V_{r1}, V_{r2}$ are the radial velocities in the azimuthal angle adjacent to $V_r$ on the equidistant circle.(Combining the position of the centroid of the squall line, using the velocity data obtained beforehand, and the maximum moving speed of the squall

line is considered to be the maximum wind field velocity. The squall lines in different layers of the scanning data whose centroid converges within a distance (R) are considered to be the same squall line. (The process of calculating R takes into account the fact that the squall line's shape might change in the process of moving and evolution process which results in the centroid shifting, so the calculate method introduces the width of the squall line, which improves the searching ability).

$$R = V_{max} * \Delta time + width(SQL)/2 \tag{8}$$

where $V_{max}$ is the maximum wind field velocity obtained by inversion in the squall line region, $\Delta time$ is the scanning data

time interval, and $width(SQL)$ is the squall line width estimated above.

Applying the above method to data with different elevation angles from the same volume-scanning process, the three-dimensional structure of the squall line can be obtained. Applying it to data from different volume-scanning processes allows the squall line to be tracked.

The overall flowchart of the algorithm is as follows:

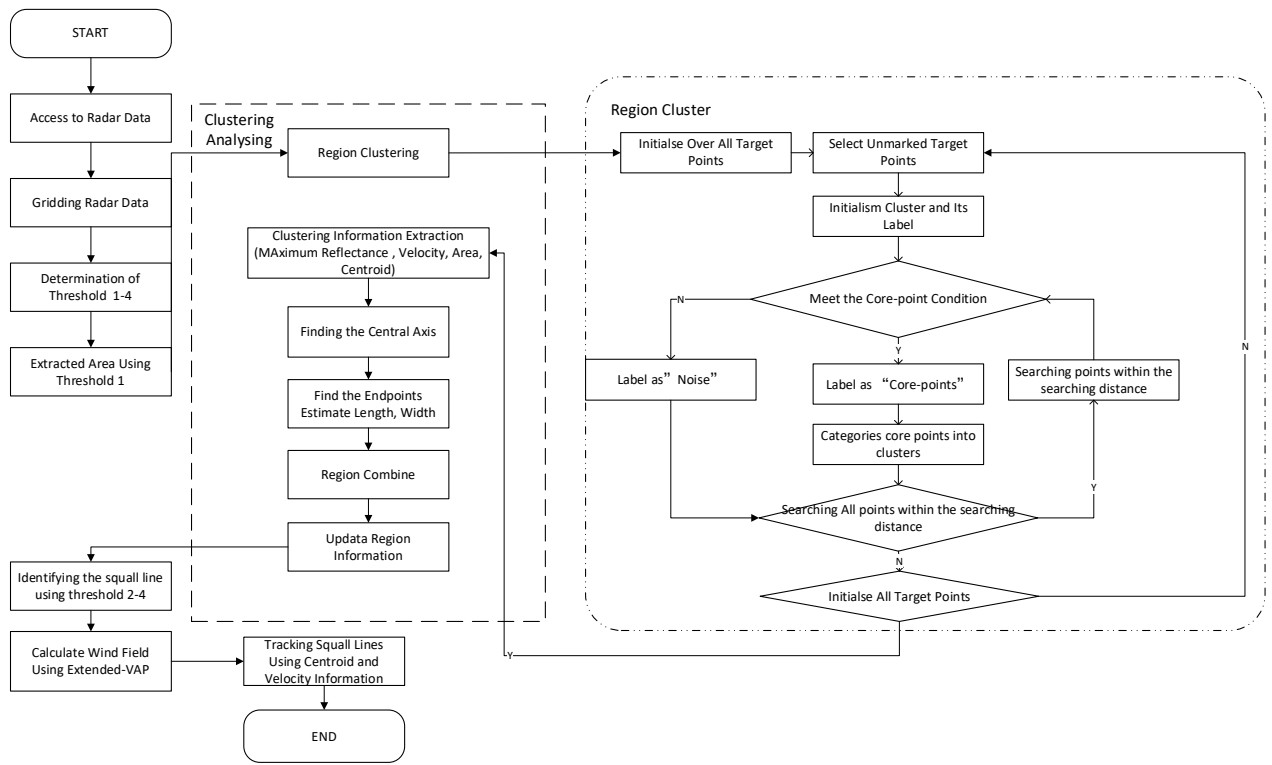

**Figure 3 Overall flowchart**

**3 Results**





## 3.1 Experimental design

The method is based on NEXRAD and CINRAD, thus, in order to test the efficacy of this method, radar data from both weather radar networks are employed for the experiment.

A typical squall line was observed by the CINRAD Z9762 weather radar in He Yuan, Guangdong Province, on June 4, 2016. Partial thunderstorms and gusty weather occur that day. The radar recorded the developmental stage and exuberant stage process of the squall line. Through visual observation by meteorologists, a strong echo band with a length of about 200km was found in the radar echo image. The volume-scan data of the radar on that day is selected to demonstrate the algorithm-identifying process and show the results of each step in detail. Meanwhile, the demonstration of the three-dimensional structure merging and dynamic tracking is shown. Based on this example, the anti-interference ability is verified by adding artificial noise interference as well.

In order to ensure good applicability of the algorithm, the ability to identify more than one squall line at the same time needs to be taken into account. A special squall line process was observed in the United States on July 1, 2014. The hurricane caused a large area of severe wind damage. Two squall lines were sounded in the same volume-scan data. Observations from the Chicago-based KLOT radar were chosen to validate the algorithm's ability to identify, 3D merge, and track multiple storm lines in the presence of multiple squall lines at the same moment in the volumetric scan data.

In order to be closer to the actual situation, and to verify the performance of the method objectively, the radar volume scanning data related to tornadoes are selected to verify the algorithm's performance in the identification process by comparing the algorithm identification with the manual identification results.

## 3.2 Example from Z9762 in HeYuan, Guangdong China

### 3.2.1. Static identifying

Static identifying means the process of identifying in single layer of radar volume scan data. The third layer (data with an elevation angle of 1.36) of the volume scan data of the Z9762 radar in Heyuan City, Guangdong Province, China, in 2016, at 7:00:00 UTC, is selected as a typical example to analyses the identifying capability of the algorithm in a single-layer of volume scan data.

Firstly, the radar data are gridded, and the nearest neighbor interpolation method is used to obtain the information of the echo data in real space (shown in Figure 4 (a)).

Using threshold 1, the points with a reflectivity factor that are significantly higher than the surrounding area are extracted as shown in Figure 4 (b)

shows the points extracted using threshold 1, a squall line can be visually found in the image, but it should be noticed that there are also some points that do not correspond to squall lines, further analysis is needed to extract the squall line from these points.





The above data points can be classified based on the point's density and searching distance by the clustering method. In the
clustering method, the density clustering parameters are set as follows: Eps = 5, MinPts = 30, Searching Distance = 7 (The
parameters are sited to differentiate the storm cells with large areas and the isolated points. Meanwhile, it is not sensitive to
the threshold). The classification results obtained in the Region clustering step of the clustering analysis are shown in Figure
4 (c).

Cluster characterization is based on the clusters in the above results. Using the cluster labeled as 1 to show the detail of the
Region characterization step. The cluster's area is calculated to be 1,239 km$^2$. The central axis of the cluster obtained through
the Hough transform, and the two endpoints are shown in  Figure 4 (d), The cluster is estimated to have a length of 120.7km
and a width of 15.4km. The central intensity of the cluster is 61dBZ.

Iterate over all the clusters and merge the satisfied clusters by their endpoints' distance. Get the result shown in Figure 4 (e)
Re-calculate the clusters' features, and verify whether they meet the threshold 2-4 (mentioned in Table 2). The cluster that
meets the thresholds is considered to be the squall lines (Figure 4(f)).

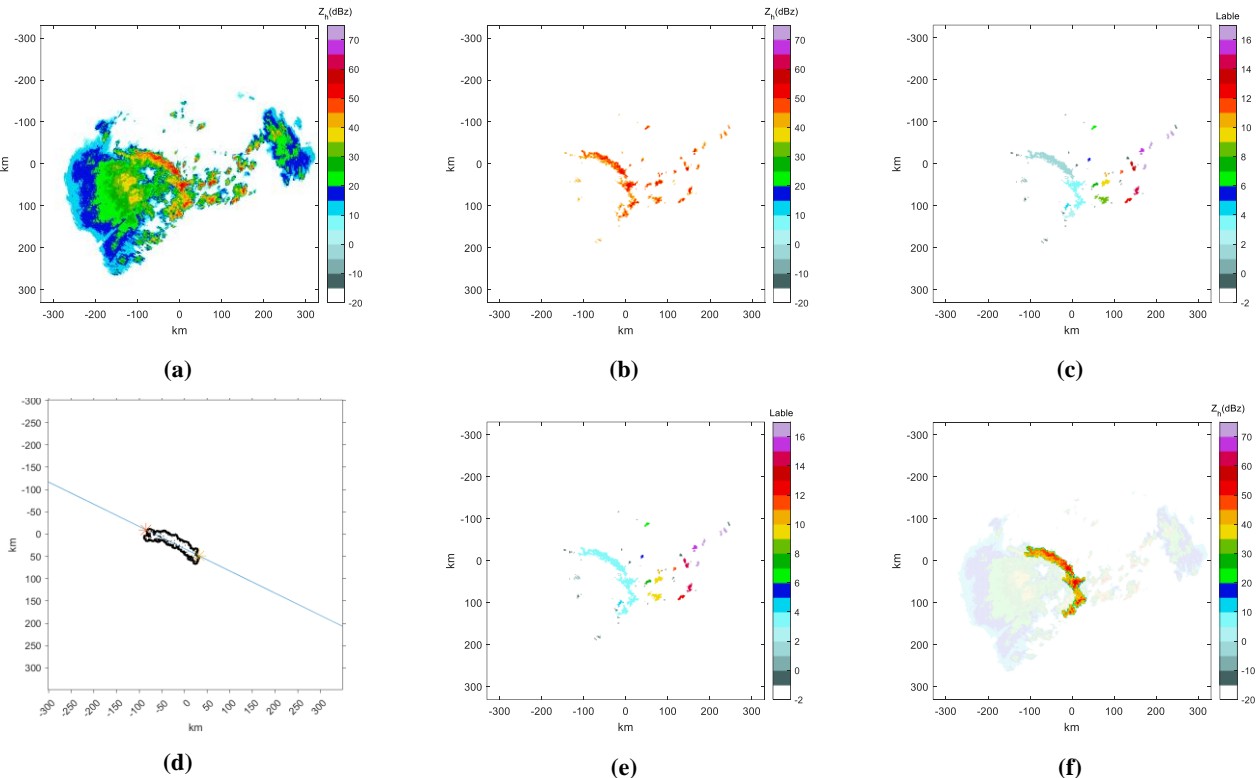

**Figure 4 (a) The weather system detected by Z9762 7:00:00 UTC Jun 04, 2016, with a squall line that can be seen in the figure.
(b)The points extracted using the threshold, the reflectivity of these points is significantly higher than the surrounding point. (c) The
results of region clustering, the colorbar indicate different clusters.  (d) An example of central axis obtaining. The edge of the cluster
and the identified central axis, and the endpoints of the obtained cluster central axis are marked as '*', and the length of the cluster**





**can be calculated using these two points. (e) The results of region combination. (f) Shows the result of squall line identification (Areas with a hundredth of white opacity are identified as squall lines).**

The above result shows that the algorithm is effective in identifying typical squall lines in single-layer radar data, and can
effectively identify the existence of squall lines and mark the location of squall lines, as well as differentiate squall lines between convective cells and clutter points.

### 3.2.2. Anti-interference capability

This algorithm in principle uses quality-controlled weather radar data. However, as mentioned in the prior study (Wang et al., 2021), for the case of some interference, the traditional algorithm is unusable. However, in the actual use of the method, there
will be interference that may not be eliminated in the quality-control process with a small probability of occurrence. Compared to the traditional method, this method has a higher degree of interference resistance. Co-channel interference data are added to the reflectivity data by random replacement or addition to test the anti-interference performance of the algorithm, and the simulated interference and the identification of the results are shown .

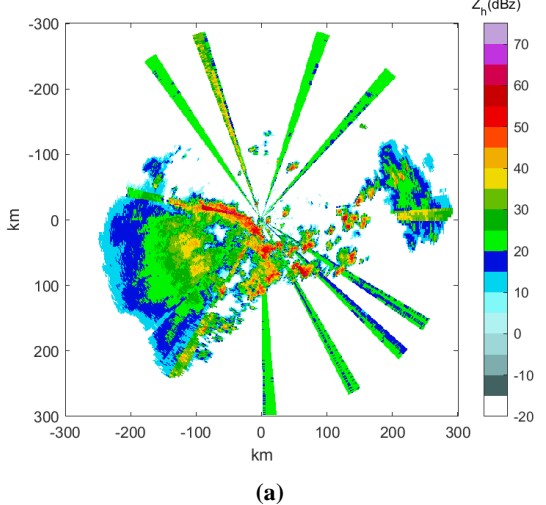

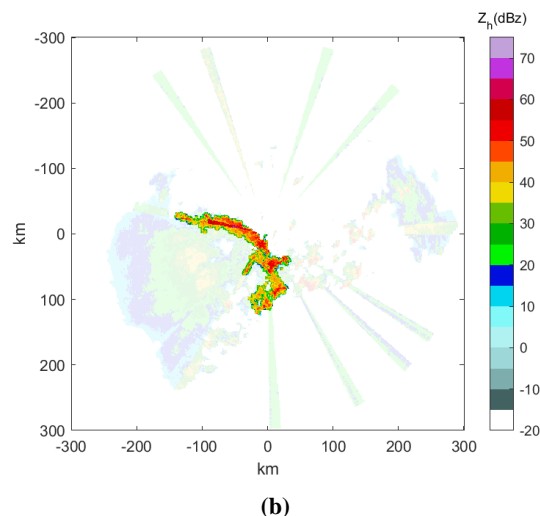

               **(a)**                       **(b)**

**Figure 5 The radar data with random co-channel noise added (a) and the result of the squall line identifying the squall line (b)**

The results show that this method is more robust than the traditional method in the process of static identification when encountering interference. The method can identify the squall line in the data with co-channel interference if the interference doesn't cover the weather information, while the method in the prior study is impossible to effectively identify.



### 3.2.3 Three-dimensional structures

The 3D structure of the squall line obtained by merging the different layers of volume-scan data using centroid and EVAP is shown in Figure 10:

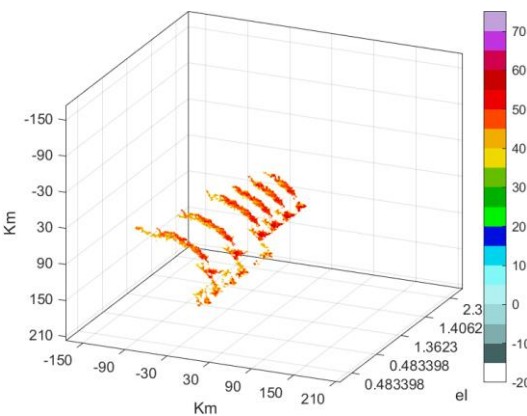

**Figure 6 The squall line's 3D strut of the reflectivity data.**

### 3.2.4 Target racking

The results of the squall line tracking using the above method are as follows (the third layer of volume-scan data for the period 05:54:00-07:00:00 is chosen to demonstrate the tracking effect).

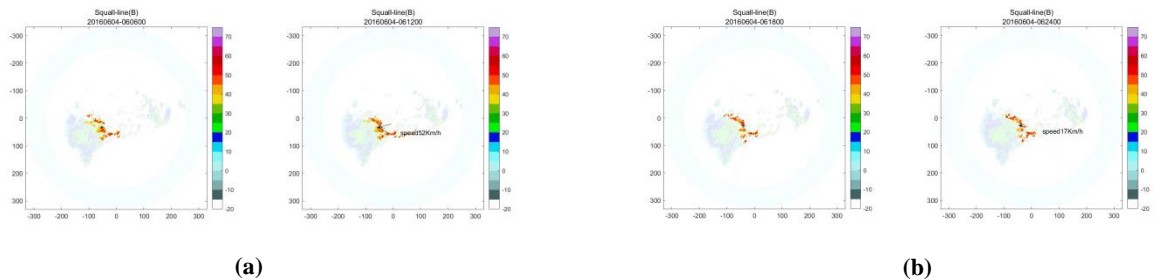

(a)                                                                                    (b)

**Figure 7 The result of squall line tracking of 6:06:00 UTC Jun 04, 2016 (a), and6:18:00 UTC Jun 04, 2016 (b). The figure shows the squall lines' movement of the latter moment relative to the former moment. The '*' shows the location of the centroid of the squall lines, and the arrows show the moving direction of the squall lines. The moving speed of the squall lines is given on the right side of the arrow.**

The results show that the algorithm can effectively track squall lines in the development and exuberant stages of the squall line.

### 3.3 Example from KLOT in the US

### 3.3.1. Static identifying

The static identification result of the KLOT's lowest elevation level III data observed at 01:13:00 UTC Jul 01, 2017, is shown in Figure 8 . Two squall lines have been identified by the algorithm.





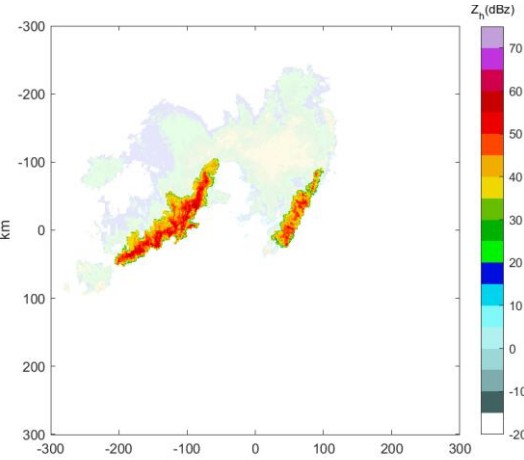

**Figure 8 The result of squall line identification on KLOT at 01:13:00. UTC July 01, 2014. (Areas with a hundredth of white opacity**
**are identified as squall lines)**

### 3.3.2. Three-dimensional structures

The volume scan data from KLOT on 01:16:55 UTC Jul 01, 2017 level II data was selected for the validation of the radar data

3D merging capability of the double squall line process, and the 3D structure is shown in Figure 13.

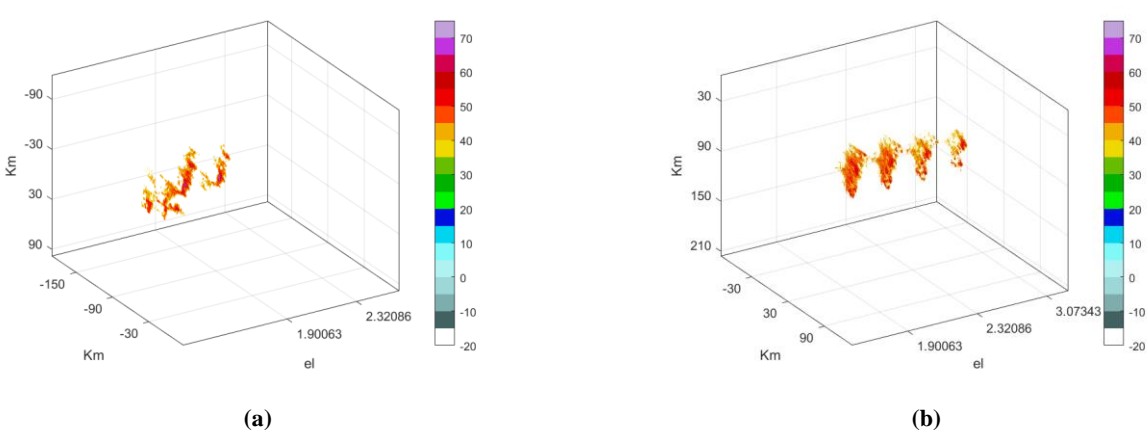

**Figure 9 The 3D strut of the reflectivity data of the squall line on the left in the radar image (a) and the squall line on the right (b).**

### 3.3.3  Target racking

The two squall lines appearing in the above examples are tracked separately, and the tracking results are respectively shown
in Figure 10





The above results show that when two squall lines appear in the volume scanning process of the same radar, the algorithm is able to track the two squall lines separately. At the same time, the method can give the direction and the speed of movement.

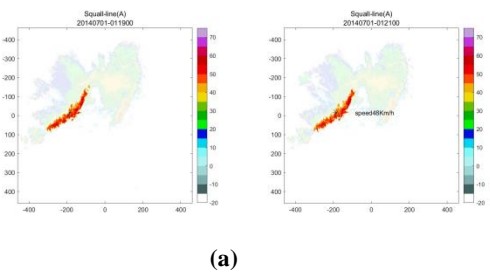

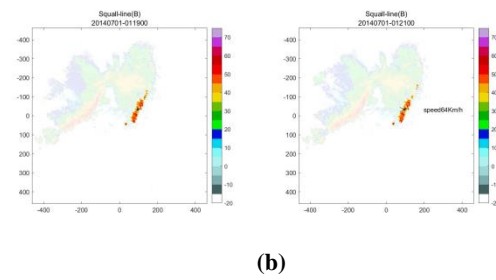

(a)                                                                 (b)

**Figure 10 The result of tracking for the two squall lines on 01:13:00 UTC Jul 01, 2017 are shown separately in (a) (b), the figure shows the squall lines' movement of the latter moment relative to the former moment. The '*' shows the location of the centroid of the squall lines, and the arrows show the moving direction of the squall lines. The moving speed of the squall lines is given on the right side of the arrow.**

### 3.4 Quantitative analysis

In order to be closer to the actual situation and do more objective verification of the performance of the algorithm, a large amount of verifying experiments is needed. Due to the limited observational range of a single radar, the appearance of squall lines is characterized by randomness. The number of available radar data is limited in finding data containing squall lines. There is a certain connection between the existence of squall lines and the occurrence of tornadoes. The weather radar data associated with tornadoes observed in the Jiangsu province (The data selected are shown in Table 3) are selected as the dataset of quantitative analysis. Precipitation data from 2022/11 to 2023/05 obtained from the RXM25 radar in Chengdu were also selected. The performance of the algorithm was verified using two approaches, one using manually identifying data to compare with the algorithm results and the other using the TITAN algorithm identifying results.

Comparing the results of manual identification or TITAN with algorithmic identification, the confusion matrix is obtained as follows. The results are shown in. In manual identification, firstly use computers to search out the data to meet the following request: The highest reflectivity is greater than 50dBZ, and the number of points greater than 40dBZ is not less than 1,000. Then manually select the reflectivity data and meet the following request: There is a region in the radar-girded data with a reflectivity of not less than 40dBZ, the area of this region is not less than 2000Km$^2$ and the length is not less than 100Km, and the maximum reflectivity of the region is more than 50dBZ. The data thus selected is considered to be manually identified squall lines.

The following events are defined by taking the manual (or TITAN) identifying result as the true result of the sample (If a squall line is manually observed in the radar echo image, the squall line is considered to actually exist within the radar observation range) in the quantitative analysis process: True Positive (TP): the true result of the sample is positive, the algorithm predicts result is positive. True Negative (TN): the true result of the sample is negative, and the algorithm predicts the result is negative.





False Positive (FP): the true result of the sample is negative and the algorithm predicts the result is positive. False Negative (FN): the true result of the sample is positive and the algorithm predicts the result is negative. Based on the above samples, the

350 following parameters are defined to reflect the algorithm performance.

| Radar Name | Observation time | Radar Name | Observation time |
|---|---|---|---|
| Z9250 | 2007/07/03 | Z9517 | 2016/06/23 |
| | 2011/07/12 | | 2017/08/01 |
| Z9513 | 2009/08/27 | Z9518 | 2008/07/04 |
| | 2011/07/13 | Z9516 | 2006/07/03 |
| | 2016/07/06 | | 2008/07/29 |
| Z9515 | 2006/07/03 | | 2008/07/30 |
| | 2008/07/29 | | 2008/08/17 |
| | 2008/07/30 | | 2011/07/11 |
| | 2008/08/17 | | 2012/08/10 |
| | 2011/07/11 | Z9519 | 2011/08/02 |
| | 2012/08/10 | | 2016/07/06 |
| Z9523 | 2013/07/07 | Z9527 | 2017/08/01 |
| | 2015/07/24 | | 2018/08/18 |
| | 2017/07/02 | | |

**Table 3 The data source information**

| Algorithm \ Manual | Y | N |
|---|---|---|
| Y | 1040 | 99 |
| N | 54 | 7409 |

'Y' means there are squall lines identified, 'N' means there are no squall lines identified

**Table 4 The Manual identifying and algorithmic identifying results**

| Algorithm \ TITAN | Y | N |
|---|---|---|
| Y | 953 | 186 |
| N | 46 | 7417 |

'Y' means there are squall lines identified, 'N' means there are no squall lines identified

**Table 5 The TITAN identifying and algorithmic identifying results**

Accuracy of algorithmic recognition:

$$ACC = {TP + TN}/{TP + FP + TN + FN} \qquad (9)$$

Successful squall line identifying the rate of the algorithm:





$$PRE = {TP}/{TP + FP} \tag{10}$$

False identifying  rate of the algorithm:

$$FAR = {FN}/{TN + FN} \tag{11}$$

Missed identifying rate of the algorithm:

$$NAR = {FP}/{TP + FP} \tag{12}$$

In this method, take the manual identifying result as the true result, the calculated result is that: $ACC = 98.25\%$ , $PRE =$ 95.06% , $FAR = 3.17\%$ , $NAR = 4.93\%$. Meanwhile take the TITAN identifying result as the true result, the calculated result is that: $ACC = 97.31\%$ , $PRE = 95.40\%$ , $FAR = 16.33\%$ , $NAR = 4.84\%$.

The above tests show that the accuracy and recognition rate of this method is higher than 95% . Using the manual identifying
results as a benchmark, the FAR and NAR than 5%. However, the FAR is over 15% when using the TITAN result as a benchmark. Comparing the TITAN with the manual identifying result, we found that TITAN identified the liner-arranged storm cells as multiple independent cells during the squall line development stage.

The TITAN has been utilized in operations, following Figure 11 shows the products generated by the TITAN algorithm in the operations of the CINRADSA radar network. The product shows the squall line identifying the result of Z9762 from 6:06:00
to 6:24 UTC on Jun 04, 2016 (Figure 11(a)-(f)). The product shows the storm cells' location by the centroid and uses an arrow to show the movement of the cells. The squall lines also can be displayed separately in an independent product. The identifying results during this time period of the method designed in this experiment are shown in Figure 11 (g) –(i)

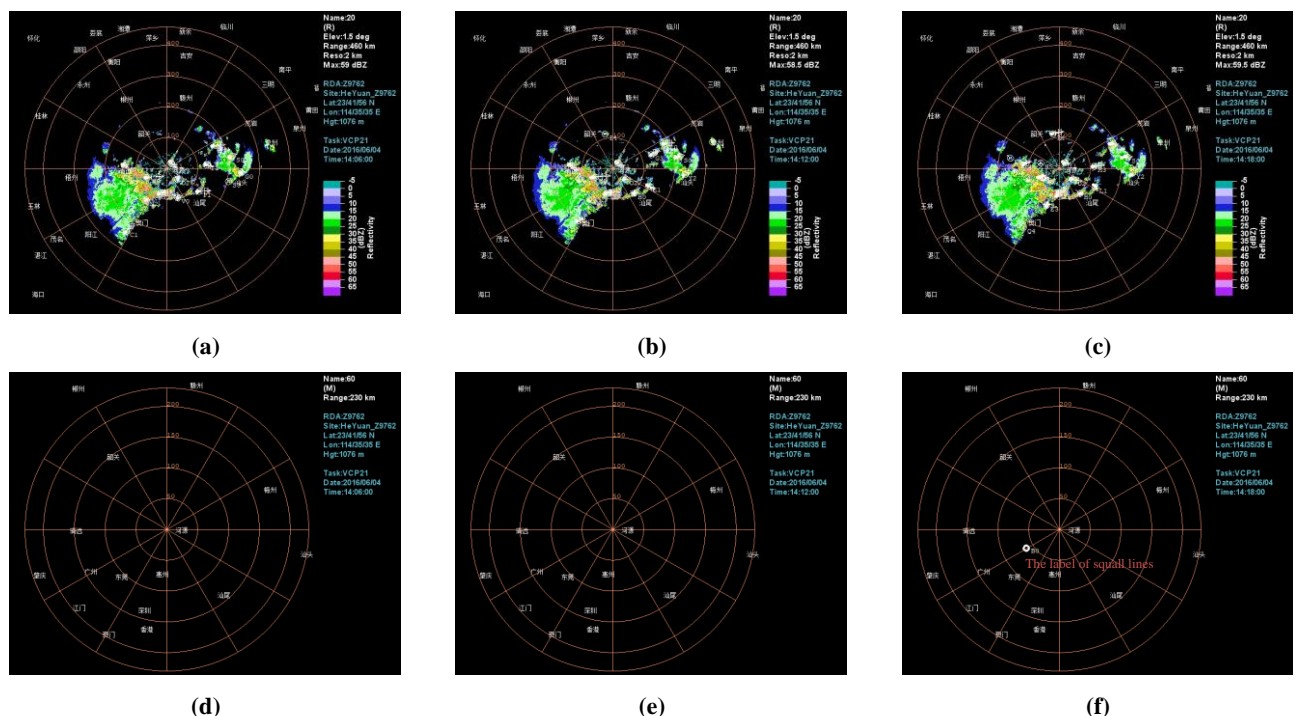



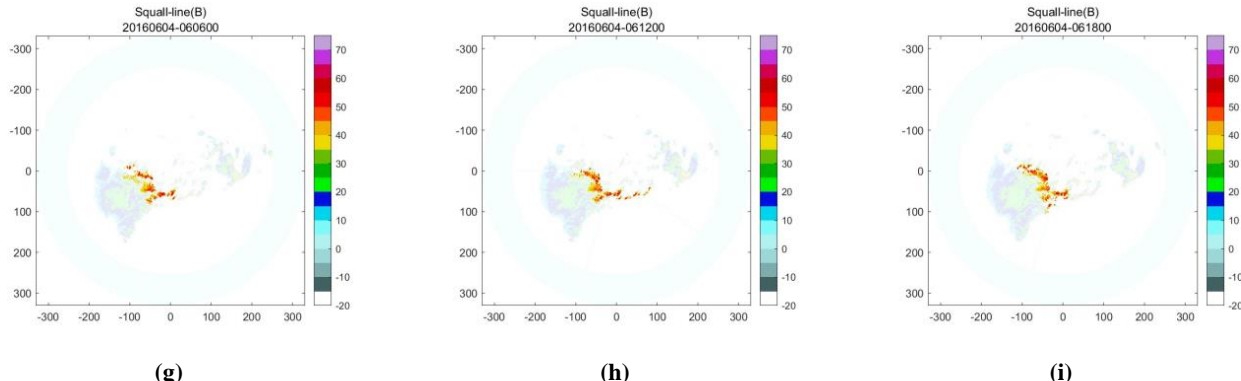

|                | **(g)**                | **(h)**                | **(i)**                |

**Figure 11 (a)-(f)The products generated by the TITAN algorithm. (g)-(i) The identifying result of the method in this study. The figures (a)-(c) show TITAN's identifying result of all storm cells on 6:06:00 UTC Jun 04, 2016UTC(a), 6:12:00 UTC Jun 04, 2016UTC(b), 6:18:00 UTC Jun 04, 2016UTC(c). The TITAN's mesoscale convection identifying results or the corresponding moments of (a) (b) (c) are specially labelled in (d)-(f), and the label is near the text in (f). The bottom three figures (g)- (i) show the identifying results for the corresponding moments of the method designed in this study. The time shown in the (a)-(f) refers to Beijing time (UTC +8)**

As the result shown in the Figure 11, the method in this study firstly identifies the squall line at 6:06:00 UTC, while the TITAN identify the squall line at 6:18:00 UTC. Through the above figures, it is easy to find that when the cells not fully merged TITAN identify them as independent cells. The method of this study identifies squall lines before they are fully merged. It is demonstrated that the squall warning can be improved compared to the TITAN algorithm. By statistically analyzing the results with manual identification, the algorithm is able to advance the squall line warning time by about 15 minutes. This suggests that the combination of storm cells in a linear arrangement does allow the method to early identify the squall lines.

Overall, this method can effectively identify the squall line in the selected weather radar data. This method also can effectively identify the squall line in the development stage and exuberant stage. The method can also provide the three-dimensional structure of the squall line and track the squall line before it is fully merged. The algorithm can improve the timeliness of the weather forecasting process by the above advantages.

## 4 Conclusion and discussion

In this paper, an automatic squall line identification and tracking algorithm for weather radar echo data is proposed. Taking the Doppler weather radar data as the data source. The points that are significantly higher than the surrounding area are extracted by threshold from the pre-processed radar data. The points are distinguished into clusters by the clustering method. The clusters' features including reflectivity value, length, width, area, endpoints, central axis, and centroid are obtained by the clustering characterization. The clusters linearly arranged are merged to improve the identifying ability in the development stage of squall lines. The movement tracking and three-dimensional structure of the squall line are obtained using the centroid and velocity of the squall lines identified in a single layer.



By analyzing two examples from two different radars, it was proved that the method is able to identify one or more squall lines in the radar data effectively. In the process of quantitative analysis of the algorithms, the manual identification or TITAN identification results were identified as true samples. Both analyses show that the method has a high accuracy rate. The analysis of the TITAN results shows that the method in this study is able to advance the early warning of squall lines. However, the manual identification process is still somewhat subjective, therefore further optimization experiments using other polarisation parameters, data from weather stations, etc. are required.

Compared with the traditional method, this method does not rely on manual observation, so the identification and tracking process can be automated through the computer to improve the accuracy and timeliness of weather warning operations.

Using this algorithm together with other convective identification and tracking algorithms, the information of squall lines, storm cells, supercells, and other targets is used simultaneously, and the prediction ability for catastrophic weather including tornadoes will be greatly improved, and combined with the traditional weather warning algorithms, it can further improve the reliability of catastrophic weather warning work. A finer vertical structure of squall may be obtained with the deep-learning technology and the 3D structure obtained in this method.

*Author Contributions:* Conceptualization, Y.W., Z.S., and J.H.; methodology, Z.S. and Y.W.; software, J.H.,. and Z.S.; formal analysis, Z.S., Y.W. and J.H.; writing—original draft preparation, Y.W.; writing—review and editing, J.H., Y.W.; visualization, Y.W.. All authors have read and agreed to the published version of the manuscript.

*Funding:* This work was supported by the National Key R&D Program of China (2021YFC3090203), the Key Laboratory of Atmospheric Sounding Program of China Meteorological Administration (U2021Z01, U2021Z09), and CMA Meteorological Observation Centre (CMAJBGS202203)

*Data Availability Statement:* Not applicable.

*Acknowledgments:* The authors would like to express their sincere thanks to Guangdong Meteorological Network and Equipment Support Centre for supplying the data used in this manuscript and their viewers for their constructive comments and editorial suggestions that considerably helped improve the quality of the manuscript.

*Conflicts of Interest:* The authors declare no conflict of interest.

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
