# Peer review of "A clustering-based method for identifying and tracking squall line"

_Atmospheric Measurement Techniques, 2023_

## Author Response (AR1)

RC1:

1. The describe of the methods are relatively vague, especially the connection between the five steps. What is the purpose of each step?

Dear Editor, thanks for your kind comment. A paragraph summarizing the methodology has been added at the beginning of the section and optimized the description based on your kind comment.

Added paragraph: The squall line carries a large number of precipitation particles, so its reflectivity factor is significantly greater than that of the surrounding area. The spatial characteristics of the squall line are that the convective system is linearly distributed and covers a large area, which makes it appear in the radar image as a high reflectivity echo band with a large area and long length. This algorithm uses the spatial and temporal evolution characteristics of squall line echoes to extract points with reflectivities that are significantly greater than those in the surrounding area, and through the distribution of points, the noise is filtered out, and the points are divided into clusters. The areas that met the spatial structural characteristics of squall lines were filtered out, and the results of squall line identification were ultimately obtained.

2. How are the three parameters set in the step of region clustering? Eps, MinpPts and Searching Distance

Dear Editor, thanks for your kind comment. We describe the setting of parameters in detail and add it to the corresponding section.

Determine three parameters: Eps, MinPts, and the searching distance. According to Orlanski's classification of convective scales(Orlanski, 1975), squall lines are $\beta$-size convective systems; however, in the development stage, squall lines consist of several smaller scale convective systems ($\gamma$-size convective systems with a range of 2-20 km) in a linear and tightly packed formation. When searching the points using the Eps range, the presence of $\gamma$-scale convection cannot be neglected, so Eps should be in the range of 1-10 km in this algorithm (Eps is insensitive to the threshold in this range). Missing radar data usually occur in 1-2 radial data points, and the searching distance and EPS take the following values:

Searching Distance=EPS+2×RadialGate     (2)

where RadialGate is the length of the weather radar radial gate. The MinPts is set to the maximum area of the isolated point or noise to be detected. In a previous study (Wang et al., 2021), the number of valid points present in a rectangular box of n*n (the number of valid points threshold is 0.25× number of the point) was used to determine the presence of isolated points, so the MinPts was approximately 0.25 ×, which is the area of the circle with the searching distance as the radius (round towards negative infinity and retaining one valid digit).

3. Some expressions in the text need to be polished.

   Dear Editor, thanks for your kind comment. Sentences in this article have been polished.

RC2:

1. The methodology section provides a detailed description of the proposed method, yet it lacks a comprehensive discussion on the selection criteria for key parameters in the clustering algorithm. Clarifying how parameters like the radius of the field (Eps), minimum number of points (MinPts), and searching distance were optimized would enhance the reproducibility and robustness of the findings. The authors claimed that "it is not sensitive to the threshold" in line 257; the manuscript would benefit from a sensitivity analysis demonstrating how variations in these parameters affect the identification and tracking outcomes.

Dear Editor, thanks for your kind comment. We realized that the previous manuscripts were not clear about the parameter settings, which led to some misunderstandings. So, we describe the setting of parameters in detail and add it to the corresponding section.

Determine three parameters: Eps, MinPts, and the searching distance. According to Orlanski's classification of convective scales(Orlanski, 1975), squall lines are β-size convective systems; however, in the development stage, squall lines consist of several smaller scale convective systems (γ-size convective systems with a range of 2-20 km) in a linear and tightly packed formation. When searching the points using the Eps range, the presence of γ-scale convection cannot be neglected, so Eps should be in the range of 1-10 km in this algorithm (Eps is insensitive to the threshold in this range). Missing radar data usually occur in 1-2 radial data points, and the searching distance and EPS take the following values:

Searching Distance=EPS+2×RadialGate     (2)

where RadialGate is the length of the weather radar radial gate. The MinPts is set to the maximum area of the isolated point or noise to be detected. In a previous study (Wang et al., 2021), the number of valid points present in a rectangular box of n*n (the number of valid points threshold is 0.25× number of the point) was used to determine the presence of isolated points, so the MinPts was approximately 0.25 ×, which is the area of the circle with the searching distance as the radius (round towards negative infinity and retaining one valid digit).

2. The manuscript talked about the anti-interference capability of the developed method. Can the authors add some discussion about to what extent does the quality of radar data (e.g., resolution, coverage, preprocessing techniques) affect the performance of the clustering-based method? And how does this impact compare with the sensitivity of other squall line identification methods to data quality?

Dear Editor, thanks for your kind comment.

The reason for mentioning interference immunity in this method is that in the previous study of squall lines using Hough Change, it was mentioned that it would unable work properly in the disturbances mentioned in the text, and in fact, the previous method would recognize the disturbances as squall lines(as shown in the figure below). However, this method can work more effectively as shown in the manuscript.

[Figure]

3. The practical implications of the research for improving early are discussed; however, this discussion could be expanded. Specifically, insights into how the method could be integrated into existing meteorological observation and forecasting frameworks, and the potential challenges and solutions in doing so, would be highly valuable.

    Dear Editor, thanks for your kind comment.

    This method can identify the squall lines earlier than the traditional method. The squall lines are often associated with tornadoes, downburst storms, and other catastrophic weather events, and the earlier identification of squall lines in the early warning system can be used to send out warning information of the related meteorological events. Meanwhile, in the usage of the collaborative radar system, combined with the identification of squall lines, refined structural detection results will be carried out and more accurate analysis results will be derived, leading to more precise warning results.

4. The manuscript would benefit from a more detailed examination of its limitations. For instance, the impact of radar data quality and coverage, potential biases, and the method's performance under extreme weather conditions are areas that warrant further discussion. Suggestions for future research directions, possibly including the integration of machine learning techniques for parameter optimization or the exploration of the method's application to other types of convective weather systems, would be beneficial.

Dear Editor, thanks for your kind comment.

The identification of the squall line in this method is mainly based on the radar reflectivity data, which may not be very accurate for the edge marking of the squall lines. Therefore, other radar parameters will be used in combination with machine learning algorithms to obtain accurate edges of the squall lines in subsequent studies. Another limitation is that the identification and tracking process of this method only works in the scanning results of a single radar, which requires the radar to scan the complete squall lines, and in the subsequent research, the identification of squall lines in a larger coverage will be realized by the girded data of multiple radars.

5. The font size of the text (axis ticks and labels) are too small, could you please make the figures clearer?

Dear Editor, thanks for your kind comment.

The figures have been resized in the manuscript.

6. Lines 99-100, "The method is based on volume scan data from NEXRAD and CINRAD. The method is based on the weather radars that are already in operational usage." Please rephrase the sentences.

Dear Editor, thanks for your kind comment.

This sentence has been rephrased in the manuscript.

7. Line 292, 313, figure number errors.

Dear Editor, thanks for your kind comment.

This error has been corrected in the manuscript.